# Establishment of a Wisent (*Bison bonasus*) Germplasm Bank

**DOI:** 10.3390/ani12101239

**Published:** 2022-05-11

**Authors:** Anna Maria Duszewska, Magdalena Baraniewicz-Kołek, Jarosław Wojdan, Katarzyna Barłowska, Wojciech Bielecki, Paweł Gręda, Wojciech Niżański, Wanda Olech

**Affiliations:** 1Department of Morphological Sciences, Faculty of Veterinary Medicine, Warsaw University of Life Sciences, Nowoursynowska 159, 02-776 Warsaw, Poland; magdalena_baraniewicz_kolek@sggw.edu.pl (M.B.-K.); pawel.greda@sggw.edu.pl (P.G.); 2Department of Biotechnology and Nutrigenomics, Institute of Genetics and Animal Biotechnology, Polish Academy of Sciences, ul. Postępu 36A, Jastrzębiec, 05-552 Magdalenka, Poland; jawo4@op.pl (J.W.); k.barlowska@igbzpan.pl (K.B.); 3Department of Pathology and Veterinary Diagnostics, Faculty of Veterinary Medicine, Warsaw University of Life Sciences, Nowoursynowska 159, 02-776 Warsaw, Poland; wojciech_bielecki@sggw.edu.pl; 4Department of Reproduction and Clinic of Farm Animals, Faculty of Veterinary Medicine, Wroclaw University of Environmental and Life Sciences, 50-366 Wroclaw, Poland; wojciech.nizanski@upwr.edu.pl; 5Department of Genetics and Animal Breeding, Faculty of Animal Science, Warsaw University of Life Sciences, 02-776 Warsaw, Poland; wanda_olech_piasecka@sggw.edu.pl

**Keywords:** wisent, European bison, oocytes, sperm, embryos, embryo transfer, germplasm bank, threatened species

## Abstract

**Simple Summary:**

The wisent (European bison) is a protected species. For this reason, we undertook the use of biotechnologies—such as in vitro maturation of oocytes, in vitro fertilization of matured oocytes, in vitro culture of embryos, and embryo vitrification—to establish a wisent embryo bank. The competencies of the vitrified embryos were tested by transferring the warming embryos to cattle (interspecies embryo transfer). The pregnancy was confirmed biochemically and using USG, and although the fetuses were resorbed, the embryos’ competence for development was demonstrated. The results of these studies open the way for the cryoconservation of wisent germplasm.

**Abstract:**

The wisent, or European bison (*Bison bonasus*), belongs to the same family (Bovidae) as the American bison and domestic cattle. The wisent is the largest mammal in Europe, and is called the “Forest Emperor”. The wisent is listed as “Vulnerable” on the IUCN Red List, and is protected by international law. Achievements in reproductive biotechnology have opened new possibilities for the cryoconservation of the wisent germplasm. Therefore, this research aimed to improve a strategy for the protection and preservation of the European bison through the creation of a wisent germplasm bank, based on the following procedures: isolation and in vitro maturation (IVM) of oocytes, in vitro fertilization (IVF) of matured oocytes, in vitro embryo culture (IVC), and embryo cryopreservation. Wisent ovaries were isolated from females outside the reproductive season, and eliminated from breeding for reasons other than infertility. Cumulus–oocyte complexes (COCs) were isolated from follicles greater than 2 mm in diameter and matured for 24 h and 30 h. After IVM, COCs were fertilized in vitro with wisent sperm. The obtained wisent zygotes, based on oocytes matured for 24 h and 30 h, were cultured for 216 h. Embryos at the morula and early blastocyst stages were vitrified and then warmed and transferred to interspecies recipients (*Bos taurus*). USG and biochemical tests were used to monitor pregnancies. This study obtained embryos in the morula and early blastocyst stages only after oocytes were fertilized and matured for 30 h. On average, per oocyte donor, 12.33 ± 0.5 COCs were isolated, and only 9.33 ± 0.61 COCs were qualified for in vitro maturation (75.68%), while 9.16 ± 0.48 COCs were matured (84.32%). On average, per donor, 5.5 ± 0.34 embryos were cleaved (59.96%) after 48 h post-fertilization (hpf), and 3.33 ± 0.21 achieved the eight-cell stage (36.52%) after 96 hpf, while 1 ± 0.21 morula and early blastocyst stages (10.71%) were achieved after 216 hpf. A total of six embryos (one morula and five early blastocysts) were obtained and vitrified; after warming, five of them were interspecies transferred to cattle (*Bos taurus*). On day 41 after fertilization, 3 out of 5 pregnancies were detected based on USG, P4, and PAG tests. However, no pregnancy was observed on day 86 after fertilization, indicating embryo resorption. This study shows that obtaining wisent embryos in vitro, and subsequent cryopreservation to create a wisent embryo bank, can be applied and implemented for the wisent protection program.

## 1. Introduction

The wisent, or European bison (*Bison bonasus*), belongs to the same genus (*Bison*) as the American bison (*Bison bison*). Wisent can be found in the forests of Europe, and the American bison includes two subspecies: the plains American bison (*Bison bison bison*), which lives on prairies, and the wood American bison (*Bison bison athabascae*), which lives in forests. European and American bison are essential for cultural and biological reasons, and play a role as an umbrella species [1,2,3]. Currently, the European bison population in Poland is 2300, and in Europe there are 9100 individuals [4]. The main threat to the European bison is that it is a small, fragmented population with a low level of genetic diversity, which can reduce survival and fertility [5]. This increases the risks due to disease (e.g., tuberculosis, brucellosis, blindness), which can eliminate individuals or even entire herds [6,7,8,9].

The wisent is a protected species, and is listed on the IUCN Red List as “Near Threatened”. Until 2020, the European bison (*Bison bonasus*) was classified as “Vulnerable” [10,11].

Such an outstanding improvement is owed to the in situ and ex situ protection programs that have been implemented in Poland for many years [12,13,14]. The activities of this program have been supported by cryopreservation of germplasm, i.e., oocytes, spermatozoa, embryos, somatic cells, and ovarian tissues. The germplasm can be used for breeding, conservation, or research purposes. This is possible thanks to achievements in reproductive biotechnology, including procedures such as in vitro embryo production (i.e., in vitro maturation of oocytes, in vitro fertilization of matured oocytes, and in vitro culture of embryos), cryopreservation, embryo transfer, and cell and tissue cultures [15,16,17,18,19,20,21,22,23,24,25,26,27,28,29,30,31,32,33,34,35,36,37,38]. The germplasm can be cryoconserved through a slow freeze, but vitrification or ultrarapid freezing is now more commonly used. Among these techniques, vitrification deserves special attention, and is widely used by infertility clinics and institutions involved in the in vitro production of domestic and endangered animal embryos [17,18,20,21,22].

Cryopreservation of germplasm is commonly used for many species, including the American bison [23,28]. Additionally, some of these biotechniques have been implemented to protect wisents [16,32,33,36,37]. Our team has conducted this study since 2014 [39]. In 2017, we obtained the first in vitro wisent embryo [40]. In 2018, the team of Riedl et al. published an abstract about obtaining in vitro wisent embryos [41].

These successes prompted us to continue this study and focus on establishing a wisent germplasm bank, which is a multistage project defining a strategy for protecting the wisent (Figure 1).

The aims of the first step of this project were as follows: (1) to determine the conditions for oocyte maturation, fertilization, and early development of embryos in vitro; (2) assessment of the possibility of vitrification the embryos obtained in vitro; and (3) confirmation of the developmental potential of the embryos obtained in vitro by transferring them to a recipient.

## 2. Materials and Methods

This study was conducted during 2015–2021. All experiments and procedures were performed in compliance with the Polish Animal Welfare regulations, and approved by the Local Ethics Commission for Animal Experiments of Warsaw University of Life Sciences. In vitro production procedures were used according to the procedures commonly used in our laboratory [29,30].

### 2.1. Ovaries Isolation

In this study, 10 female wisents, 3–11 years old, from different herds in Poland, were culled out of the reproductive season (October–March) for reasons other than infertility and were used.

Ovaries from each donor were collected approximately 30 min after death, and were placed in DPBS containing 0.2 mg/mL streptomycin and 250 IU/mL penicillin at 30 °C, and transported in a thermos to the mobile laboratory within less than 30 min. Immature COCs were aspirated from ovarian follicles with a diameter of 2–6 mm and washed twice in the following medium: TCM 199 HEPES without NaHCO_3_, supplemented with 10% FBS (vol/vol) and 50 μg/mL gentamicin, pH 7.4. Each oocyte donor was handled individually.

### 2.2. In Vitro Maturation of Oocytes

As described by Cervantes et al. [25], before and after in vitro maturation, COCs were classified as compacted (i.e., at least one complete layer of granulosa cells tightly surrounding the oocytes), expanded (i.e., cumulus cells expanded or partially dissociated), denuded (i.e., oocytes without cumulus cells), or degenerated (i.e., with pyknotic granulosa cells and vacuolated ooplasm). Only compacted, expanded, and denuded COCs qualified for in vitro maturation. COCs from 4 donors were matured for 24 h, and COCs from 6 COC donors were matured for 30 h. Qualified COCs started maturation during transport to the laboratory in a thermos, in a modified maturation medium of TCM 199 HEPES without NaHCO_3_, supplemented with 10% FBS (vol/vol), 0.02 IU NIH-pFSH/mL, 1 μg/mL β-estradiol, 0.2 mM sodium pyruvate, 0.2 mg/mL streptomycin, and 250 IU/mL penicillin (pH 7.4), at 38.5 °C.

After maturation, COCs were analyzed as described by Cervantes et al. [25], and only expanded, compacted, and denuded COCs qualified for in vitro fertilization.

### 2.3. In Vitro Fertilization of Matured Oocytes

Only qualified COCs (i.e., expanded, compacted, and denuded) were fertilized in vitro. Sperm were previously isolated from the epididymis of a wisent bull culled out the reproductive season, and subsequently frozen in straws, as described by Kozdrowski et al. [33].

Sperm were thawed at 37 °C, centrifuged (200× *g*) for 10 min, and washed in 2 mL of Sp-TALP containing 6 mg/mL BSA fraction V and 50 μg/mL gentamicin, pH 7.4. Sperm were capacitated using the swim-up method in 1 mL of TL stock solution for sperm capacitation (MINITUBE, Delevan, CA, USA), with 6 mg/mL bovine serum albumin fraction V (BSA V) and 50 μg/mL gentamicin, pH 7.4. In vitro fertilization was performed in TL stock solution for fertilization (MINITUBE, Delevan, CA, USA), supplemented with 6 mg/mL bovine serum albumin–fatty acid fraction (BSA FAF), 0.2 mM sodium pyruvate, PHE (20 µM penicillamine, 10 µM hypotaurine, 1 µM epinephrine), 2 µg/mL heparin, and 50 μg/mL gentamicin, pH 7.4. Sperm were individually incubated with matured COCs for each donor, at a final concentration of 1 × 10^5^ sperm/oocyte, for 18 h at 38.5 °C under 5% CO^2^ in humidified air.

### 2.4. In Vitro Culture of Embryos

After 18 h post-fertilization (hpf), putative zygotes were denuded of cumulus cells by pipetting and washed twice in SOF stock solution (MINITUBE, Delevan, CA, USA) supplemented with 10% FBS (vol/vol) and 50 μg/mL gentamicin. Then, wisent zygotes were co-cultured with Vero cells (ATCC, Rockville, MD, USA) for 8–9 days. Vero cells at a concentration of 2 × 10^3^ were placed in 40 μL of SOF stock solution (MINITUBE, Delevan, CA, USA) supplemented with 10% FBS (vol/vol) and 50 μg/mL gentamicin, and overlaid with mineral oil. The wisent zygotes were individually co-cultured with Vero cells at 38.5 °C under 5% CO_2_ in humidified air until 216 h post-fertilization (hpf). At the same time, the medium was partially replaced with fresh medium at the same concentrations after 48, 96, and 144 hpf. After 168 hpf, the medium was partially replaced again, but was supplemented with 3 mg/mL BSA V instead of FBS. Embryo development was evaluated after 48, 92, and 216 hpf. Based on morphology, embryos were classified as G1, G2, or G3, based on the IETS manual [26,42].

### 2.5. Vitrification

Embryos at the morula and early blastocyst stages were vitrified using the BOVIPRO *Vit-Kit*™ (MINITUBE, Delevan, CA, USA), according to the manufacturer’s instructions, and deposited in an embryo bank. Qualified embryos for vitrification were placed in a holding medium. Each of the embryos was transferred from the holding medium to 100 μL of equilibration medium A for 5 min, and then into 100 μL of equilibration medium B for 5 min. During the second exposure, the straw was preloaded with a 90 μL column of DT medium, followed by ~1.5 cm of air. Next, the embryo was transferred to a 30 μL drop of vitrification medium and immediately loaded into the straw, followed by ~1.5 cm of air, and then another 90 μL of DT medium. The straw was then heat-sealed and placed into an empty, pre-chilled 10 mm goblet held in liquid nitrogen (LN2) vapor (−150 to −180 °C). After 1 min of vapor exposure, the goblet was lowered into liquid nitrogen (−196 °C) and the straw was submerged. The total exposure time to the vitrification media was <1 min. The straws were stored in LN2 (−196 °C).

### 2.6. Interspecies Embryo Transfer

Initially, 20 Polish black-and-white Holstein-Friesian heifers were prepared for interspecies embryo transfer. All heifers were housed in free-stall barns, fed a total mixed ration twice daily according to nutritional requirements, and had free access to water and salt licks. Two months before transfer, they were given premixed VITAMIX KW Fertility^®^ Polmass, Bydgoszcz, Poland) with A, D3, and E (6000 mg), as well as the recommended dose of beta-carotene (2000 mg) with organic selenium and biotin to improve breeding rates [43].

Estrus cycles of the recipients were synchronized by intramuscular (IM) injections of a prostaglandin F2α (PGF) analogue, administered twice at 11-day intervals. All recipients were synchronized using D-cloprostenol at a dosage of 0.15 mg/heifer (Dalmazin^®^, Fatro S.p.A., Ozzano Emilia (BO), Italy).

After the second injection of prostaglandins, all heifers were checked for signs of estrus three times per day. Heifers seen in standing estrus qualified as recipients (day 0). A day before embryo transfer, the evaluation of the corpus luteum (CL) was performed by palpations per rectum, according to Spell 2001. Only heifers with prominent CL were used for embryo transfer on day 9 (*n* = 5).

On day 9, five wisent embryos were warmed using the BOVIPRO *Vit-Kit*™ according to the manufacturer’s instructions (MINITUBE, Delevan, CA, USA). Straw with vitrified embryo was held in the air for 10 s before submersion in water at 20–24 °C for 10 s. Immediately after initial warming, the straw was held at the sealed end and flicked 3–5 times, dislodging the air pockets to mix the contents. The embryo was allowed to rehydrate via straw dilution for up to 1 min before the sealed end of the straw was cut and its contents emptied into a Petri dish. Embryos were recovered and washed using four small drops (~50 μL) of holding medium. A single warmed wisent embryo was transferred from embryo transfer medium—BioLife Transfer Medium (Agtech Inc., Kansas City, MO, USA) to the heifer uterine horn, ipsilaterally, the ovary displaying a prominent corpus luteum.

### 2.7. Monitoring Pregnancy

Recipients were monitored daily for heat behavior. Pregnancy diagnosis was made at 41 and 86 days after in vitro fertilization (dpf) by ultrasonography using an HS-2200V (Honda Electronics Co., Ltd., Toyohashi, Japan) with an intrarectal 7.5 MHz linear array transducer.

Additionally, 43 days after in vitro fertilization, progesterone (P4) and pregnancy-associated glycoprotein (PAG) serum levels were also tested to confirm pregnancy. Blood samples were collected from all recipients via puncture of a coccygeal vessel (v. caudalis mediana). The sera were separated and sent on dry ice to laboratory (LABOKLIN GMBH & CO. KG LABOR FÜR KLINISCHE DIAGNOSTIK, Bad Kissingen, Germany) for diagnosis of P4 and PAG levels.

### 2.8. Statistical Analysis

The qualified immature and matured COC rates were expressed as a ratio of the number of useable COCs to the number of recovered COCs. The rates of embryos at the 2-cell, 8-cell, morula, and blastocyst stages were expressed as a ratio of these individual stages to the number of qualified COCs for in vitro fertilization.

Statistical analyses were performed using Statgraphics Centurion (Statgraphics Technologies, Inc., The Plains, VA, USA). The effect of the time of in vitro maturation (24 h vs. 30 h) on COCs’ maturation and embryo development was analyzed using the chi-squared test. Differences at *p* < 0.05 were considered significant, and *p* < 0.01 was considered highly significant.

## 3. Results

Results of the in vitro maturation of oocytes and the development of embryos, depending on the time of oocyte maturation (24 or 30 h), are presented in Figure 1.

There were no differences between the times of in vitro maturation (24 h vs. 30 h) in terms of the percentage of matured oocytes. Significant differences were found in the percentage of embryos at the 2-cell, 8-cell, morula, and blastocyst stages (*p* < 0.01). Bars with different letters denote statistical differences at *p* < 0.01.

Oocyte maturation for only 24 h reduced the percentage of 2-cell embryos and arrested further embryonic development. In addition, none of the embryos reached the morula or early blastocyst stages, in contrast to embryos arising from oocytes that were matured for longer (30 h), which reached the morula and early blastocyst stages (10.71 ± 2.14).

Results of the in vitro maturation of oocytes and embryo development per oocyte donor, depending on the time of oocyte maturation (24 or 30 h), are presented in Table 1.

There were no differences between the times of in vitro maturation (24 h vs. 30 h) in terms of the percentage of matured oocytes (Figure 1). However, the quality of oocytes after oocyte maturation for 24 h was lower than that of those matured for 30 h (expended COCs 2 vs. 6.7—Table 1). Moreover, after 48 hpf, the development of embryos, which arose based on oocytes matured for 24 h, was lower after 24 h of oocyte maturation compared to 30 h (*p* < 0.01). The further development of these embryos was stopped. Only embryos that arose based on oocytes matured for 30 h developed until the morula or early blastocyst stages. Interestingly, one transferable embryo was obtained from each oocyte donor.

Although one embryo was obtained from each donor (six embryos in total—one morula/Grade 2; three early blastocysts/Grade 1; one early blastocyst/Grade 2; and one blastocyst/Grade 2), only five were vitrified, for technical reasons; however, all of these embryos were transferred to cattle (*Bos taurus*)—the interspecies recipients.

Figure 2 shows a wisent (European bison) immature oocyte, early blastocyst, and compact morula after warming.

The results of wisent embryo (*Bison bonasus*) interspecies transfer to cattle (*Bos taurus*) are presented in Table 2.

After the transfer, heifers were observed for the next estrus, and two of the heifers showed signs of heat. At 41 days post-fertilization (dpf), the first USG examination was performed, which revealed signs of pregnancy (i.e., enlarged horns and fluid in the uterine horns) in three heifers with the best-quality embryos transferred. The result of the USG examination was confirmed biochemically, based on progesterone (P4) and PAG tests. At 86 dpf, a subsequent USG examination was repeated, and revealed signs of embryo resorption (uterine horns returned to the state before the transfer).

## 4. Discussion

This study shows that it is possible to obtain wisent embryos in vitro, based on oocyte maturation, fertilization of matured oocytes, and embryo culture. Moreover, we found that embryos at the morula and early blastocyst stages can maintain their competence after vitrification.

### 4.1. In Vitro Maturation of Wisent Oocytes

Many factors determine the success of these biotechniques, which are crucial for access to wisent gametes. This is due to the small population of the European bison, which is also dispersed over many herds [5,16]. Moreover, gametes can only be isolated outside of the breeding season (October–March) from individuals that have been eliminated from the herd due to health reasons. In addition, gametes are isolated postmortem, which can negatively affect germplasm quality [40].

In this study, the efficiency of isolated COCs per donor (12.33 ± 0.5/12.50 ± 0.95) was similar to that for American bison [23,24,25,26,27,28]; however, analysis of COCs indicated that not all of them could qualify for maturation, which was associated with signs of degeneration (Table 1).

The main factors determining nuclear and cytoplasmic competencies for in vitro maturation may be the season, hormonal stimulation, and maturation time. In our 2018 study, we found that extending the oocyte maturation time from 24 h to 30 h impacted the development of embryos [40]. However, it is worth noting that the team of Riedl et al. [41] also conducted a similar study, and only reported that, from 2 wisents, 203 oocytes were recovered (around 102/per donor), of which 169 (83%) matured and 18 reached the morula/blastocyst stages (10.65%/per matured oocytes and 8.87% per isolated oocyte).

Much more knowledge is provided by studies on the American bison [23,24,25,26,27,28]. Krishnakumar et al. showed that immature oocytes isolated outside the breeding season and matured for 22 h have a reduced ability to fertilize, unlike oocytes isolated during the season [27]. Additionally, exciting research was conducted by Benham et al. [23], indicating that oocytes matured for 24 h reached maturity for fertilization. Thundathil et al. showed that oocytes isolated in spring—mostly from pregnant cows—and matured for 24 h were characterized by high maturity and fertilization [28]. Oocytes were also collected from hormonally stimulated donors in the breeding season. Hence, the competencies of oocytes for fertilization and embryo development were much greater, based on the work by Cervantes et al. [25,26]. The stimulation and the extension of the maturation time from 24 to 30 h—and even up to 34 h—affect oocyte maturity [26].

In our research, we also observed that oocytes cultured for 24 h mature nuclearly, which means that they achieve metaphase II. However, as confirmed by this research, they do not have sufficient fertilization and embryonic development competence, because only 18.88% of embryos reach the two-cell stage, and none reach the eight-cell stage (Table 1). This means that the embryos do not pass through a block in their in vitro development, resulting from taking control of the embryonic genome over further development [29,30].

Contrary to 24 h COC maturation, after 30 h, COCs were characterized by statistically significant dispersed cumulus (Table 1), which was a prognostic factor for oocytes’ acquisition of competencies.

An interesting observation is that, in wisents, the ooplasm is darker and unevenly distributed, which causes the yin–yang conformation effect (Figure 2A) already described in American bison [24]. Presumably, like in American bison, such a conformation does not affect the maturation of oocytes and their ability to fertilize.

### 4.2. In Vitro Fertilization of Matured Wisent Oocytes

It is worth noting that wisent sperm were also isolated outside of the reproductive season, and their ability to fertilize was tested on hybrids [44]. Furthermore, similar hybrids were also created by other researchers [35].

There seemed to be no difference in fertilization kinetics between wisent and cattle, because zygotes were formed at similar times.

### 4.3. In Vitro Culture of Wisent Embryos

In our study, embryos achieved the morula and blastocyst stages based on oocytes that matured for 30 h. On the other hand, embryos developing on oocytes matured for only 24 h only reached the two-cell stage (Figure 1). The embryos obtained based on oocytes that matured for longer (30 h) achieved the morula or early blastocyst stages, although the percentage was lower 10.71 ± 2.43% (Figure 1), but this was similar to our first report (16%) [40]. Additionally, a similar result was reported by Riedl et al., where a total of 10.65% blastocysts was achieved [41]. In American bison, the percentage of blastocysts depends on the authors; in terms of wood bison blastocysts, at day 7—27% and day 8—18.4% [26], as well as day 8—8.4% [23]. These results are also similar to ours, but other teams have achieved a higher percentage of blastocysts—45.9% (after 34 h of IVM) [24]—and morulae (32.31 ± 11.78) and blastocysts (16.42 ± 9.49) [27].

Indeed, wisent embryos’ development rate is slower than that of American bison, because the early blastocyst stage of the wisent embryos was reached in 8.5–9 days (Table 1, Figure 2B). In contrast, American bison embryos reached the blastocyst stage at the same rate as cattle embryos [31].

In our study, the extension of the culture time was not justified, as the embryos were intended for vitrification.

### 4.4. Vitrification of Wisent Embryos

Our earlier observations indicated that embryos in the morula and early blastocyst stages should be vitrified [44]. Embryos were vitrified mainly in the early blastocyst stage, i.e., before the blastocyst cavity was entirely formed. However, vitrification did not adversely affect the development of the embryos, as all of them showed normal morphology after warming (Figure 2C). Our results are similar to those of other studies [21,22].

### 4.5. Wisent Embryo Transfer to Cattle Recipients

In our study, interspecies embryo transfer was performed, and in vitro embryos were transferred to cattle (*Bos taurus*), because access to recipients was extremely limited. Additionally, Riedl et al. [41] carried out interspecies transfer of wisent embryos to *Bos taurus*. In American bison, Thundathil transferred wood bison embryos to the same species’ uterus, with successful offspring delivery in 2007 [27]. Recently, Benham et al. achieved plain bison offspring [23].

The wisent is related to *Bos taurus* and American bison, and crossbred wisent/domestic cattle and wisent/American bison are observed in nature. Hence, it was justified to check the developmental competencies of the embryos by transferring them to another species, i.e., *Bos taurus*. The interspecies transfer of embryos has been performed between many species, such as rat–mouse, donkey–horse, goat–sheep, and wild cat–domestic cat and –tiger, but only donkey–horse and wild cat embryos develop to full term [45].

Wisent embryos showed correct morphology before transfer to recipients, and developmental potential after the transfer (Table 2), which was confirmed by positive USG results at 41 dpf. This was also indicated by the levels of PAG and P4 in maternal blood. PAG is synthesized in ruminants’ trophectoderm and, after 28 days of pregnancy, can be detected in maternal blood [46]. Progesterone is a steroid hormone mainly produced by the corpus luteum, and this hormone can be detected after 28 days of pregnancy [47].

The levels of both markers indicated biochemical pregnancies in three out of five ET cases (Table 2). However, later on, based on USG, resorption took place (day 86 post-fertilization). Riedl (2018) also observed resorption after the transfer of wisent embryos to *Bos taurus* recipients [41].

Probable causes of resorption after interspecies embryo transfer have already been described by Widayati et al. [45]. These studies suggest that fetus resorption may have an immunological aspect, resulting from rejection by *Bos taurus* of a xenograft, such as a wisent fetus. There may also be an inappropriate interaction between the wisent trophoblast and *Bos taurus* endometrium, as shown by the study by Tachi and Tachi [48]. Early resorptions may also result from nutritional or support system abnormalities in vitro, becoming apparent later in embryonic development. Since we found biochemical pregnancies, it is possible that while there could be an interaction between the chorion and the endometrium, the placenta was not formed [45].

Wisent embryos could be transferred to *Bos taurus* 8.5–9 dpf at the earliest, because they only reached the stage of morulae or early blastocysts. For this reason, recipients were also between 8 and 9 days of heat. In cattle, the embryos in these stages are most often transferred to recipients on the 7th day. However, it has been shown that it is possible to successfully transfer embryos asynchronously in cattle—that is, 7-day-old or 9-day-old embryos to 9-day-old recipients [49]. Despite fetal resorption, the transfer of wisent embryos to *Bos taurus* recipients confirmed that the wisent embryos maintain developmental potential after warming.

## 5. Conclusions

In this study, the procedures of in vitro maturation of immature oocytes, in vitro fertilization of matured oocytes, and in vitro culture of embryos resulted in one wisent embryo/per donor, which could also be successfully vitrified. Therefore, these procedures allowed for the establishment of a wisent embryo bank, which can be commonly implemented to preserve and protect this species. Therefore, despite the low efficiency of this biotechnology, this is a very promising path for protecting and maintaining the genetic variability of this threatened species.

## Data Availability

No new data were created or analyzed in this study. Data sharing is not applicable.

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
