# Peer review of "Establishment of a Wisent (Bison bonasus) Germplasm Bank"

_animals, 2022, doi:10.3390/ani12101239_

Round 1

Reviewer 1 Report

General comments

The current paper is written in proper English grammar with a logical sequence and describes well the methods.

The present research aimed to improve a strategy for the protection and preservation of the European bison by development of wisent germplasm bank based on improvement of in vitro embryo production and embryo cryopreservation. Among the main achievements the authors were able to establish the proper in vitro maturation and embryo culture length. Besides, pregnancies were produced after frozen/ tawed embryos transfer of into Bos taurus recipients. However, as expected, the pregnancies did not go into term.  The novelty of present paper is the achievement of a complete program of Wisent germplasm bank, that has potential to be used among different institution all around the globe. Finally, I do recommend the paper for publishing in Animals after Minor Revisions.

Minor Revisions

Abstract

The manuscript must contain the study main achievements. It should present the main result. I strongly suggest briefly inserting the stats and results.    

Introduction

There is some missing information. Please, try to add some more literature review about the main achievements from other researchers. Bring information about oocytes and embryos production, cryopreservation. It’s important as a reader, to know the efficiency of the proposed techniques and the specie reproductive fertility.

Material and methods

Line 154- Please, briefly describe the vitrification procedure.  

Results

Figure 2: The letters in the figure are duplicated, pleach check.

Conclusions

The conclusions are consistent with the experimental design and results presented.

Author Response

Dear Reviewer

Thank you very much for your insightful review, which will improve the quality of our article.

Regarding your suggestions:

Abstract

We have attached a new abstract taking into account your suggestions

Introduction

In our opinion, we did not miss some information only to date about in vitro production of wisent embryos has limited!:

- one original article (my study from 2018)

- one abstract:

in 2018 Ridle et al. published the results in vitro production of  European bison embryos. This study was published as an abstract, but it is unavailable on the science web. Therefore I have attached it below. In this abstract, the team of Riedl et al. only reported that from 2 wisents recovered 203 oocytes (around 102/per one donor), of which 169 (83%) matured and 18 morula/blastocyst stage (11%). These data are controversial, despite I refer to them in detail in the Discussion. Also, in the Discussion, we compare the results of European bison with American bison.

Material and methods

We did it

Results

We fixed it

As suggested by another reviewer, we have made many other corrections.

Thank you very much for your engagement to help improve our article.

Sincerely,                         

Anna M. Duszewska

Anna Maria Duszewska PhD, Prof.
Division of Histology and Embryology,
Department of Morphological Sciences,
Institute of Veterinary Medicine,
Warsaw University of Life Sciences,
Nowoursynowska 159, bldg. 24, room 29
02-776 Warsaw, Poland
phone:+48225936217

Riedl, J.; Wenigerkind, H.; Weppert, M.; Wolf, E. Prospects and obstacles of in-vitro-production and interspecies transfer of European bison (Bison bonasus) embryos. International Conference „Å»ubry w dolinie Sanu”, Muczne, Poland, 05-06.09.201,  Conference Materials 2018, 72.

Prospects and obstacles of in-vitro-production and
interspecies transfer of European bison (Bison bonasus)
embryos
Johannes Riedl1, Hendrik Wenigerkind2, Myriam Weppert3,
Eckhard Wolf2,3
1 Wisentprojekt Donaumoos, Landratsamt Neuburg/Donau, Germany
2 Bayerisches Forschungszentrum für Fortpflanzungsbiologie, Badersfeld, Germany
3 Lehrstuhl für Molekulare Tierzucht und Biotechnologie, LMU München, Germany

The recovery of the European Bison after extinction in the wild is a success story
of captive breeding and re-introduction. At present more than 7000 European bison
are registered in the pedigree book. Based on the fact that recovery of the species
started with 54 animals, the population originating from 12 ancestors only is still
vulnerable from a genetic point of view. The population is highly inbred, and
the loss of genetic variability is considered detrimental to the species.
Due to the small number of animals kept in a majority of breeding centres and
the fact that captive and free-ranging herds are scattered over a vast area, animals
have to be exchanged between centres repeatedly. However, controlled exchange
of genes by transportation of individuals is laborious, costly, and implies significant
risks to the health of the selected animals.
In this context, modern technologies could be used to provide a valuable tool
to facilitate the exchange of genes between herds, and to establish a European bison
gene resource bank.
In particular, in-vitro production (IVP) of embryos and embryo transfer (ET)
offer clear benefits to conservation biology.
Transferring in-vitro produced European bison embryos to cattle recipients,
transporting them to breeding centres where they give birth to a wisent calf could
be a much easier way to exchange genetic material between distant breeding centres,
than transporting animals.
Therefore, the goal of this study was to determine the feasibility and efficiency
of in vitro embryo production and interspecies transfer to Bos taurus recipients.
Ovaries were recovered from two European Bison cows, culled in a breeding
centre to maintain a constant herd size. After culling, oocytes were collected and
subjected to a standard domestic bovine IVP procedure using frozen/thawed sperm
(Stojkovic et al., Biol Reprod 1995, 53, 1500–7). A total of 203 oocytes was retrieved
and 169 (83%) were processed for IVP, 70 (41%) of them cleaved. Among them, 18
oocytes (11%) developed to the morula (n = 8) or blastocyst (n = 10) stage. Another
17 embryos were transferred into nine synchronised domestic heifers, but no preg-
nancy was established.
In conclusion, standard protocols are feasible for IVP of European bison embryos.
However, interspecies incompatibilities between Bison bonasus and Bos taurus may
hinder widespread use of IVP

Reviewer 2 Report

 This study aimed to preserve European bison germplasm by creating a wisent embryo bank based on in vitro embryo production and embryo cryopreservation. The study concludes that bison oocytes mature after 30 h of IVM and that embryos reach the morula and early blastocyst stage between 8 and 9 days post-fertilization. Three of five recipients got pregnant after the transfer of vitrified/warmed embryos to recipients of another species (Bos taurus), although early resorptions of the fetuses were observed.

This study is very interesting and necessary to protect the European bison population. Because the use of reproductive biotechniques is extremely limited in bison species, research conducted in this manuscript is very valuable. In fact, this study is carried out only with ten female Wisent of different ages culled out of the reproductive season (October-March).

Despite its interest, this manuscript can not be accepted in the present version and some major points should be addressed.

First of all, this manuscript should be revised by an English speaker native used to correct English scientific manuscripts related to embryo technologies. It contains grammar and typo errors that should be addressed and some of the sentences are incomplete or difficult to understand.

The first objective of this work was to determine the conditions for oocyte maturation, fertilization, and early development of embryos in vitro. However only differences between 24 h and 30 h of IVM have been assessed. No differences in in vitro fertilization or culture have been considered.  The authors conclude that immature oocytes must be cultured in vitro for 30 hours to achieve maturity for fertilization. But they do not carry out any assessment of the maturation status. They conclude that 30h of IVM is better than 24 h because there are significant differences in the percentage of embryos at 2-cell, 8-cell, morula, and blastocyst stages between IVM for 24 h and 30 h. Before concluding about the better length for in vitro maturation of bison oocytes, an assessment of nuclear ( Polar body extrusion, staining of the nucleus,…) or cytoplasmic status should be included.

Besides, compact, expanded, and denuded COCs are in vitro matured. Expanded or denuded COCs should be discarded because of their lower competence to mature. Once in vitro matured,  oocytes are classified again as compacted expanded, and denuded and all of them fertilized. In this case, compacted and denuded oocytes shouldn’t be fertilized.

The discussion should include a discussion of the results. Right now it includes a new description of the results plus the results obtained by other groups without any hypothesis that would explain the differences between this work and other studies.

Change thawed for warmed throughout the paper. Vitrification implies no ice formation so it should be warmed, not thawed

Line 86: How is related “cell culture” with cryopreservation of germplasm?

Line 95: freezing >>>Vitrification

Line 111: how were the oocytes harvested from the follicles? Slicing, aspiration? How were the oocytes washed? Please describe

Line 113: Please describe Compacted, expanded and denuded COCs

Line 119: One bracket is messing and it is difficult to understand “as expanded cumulus cells expanded or partially dissociated)”

Table 1 : “6.75 [27/50] “What does the number outside the bracket means? What does the number in the bracket mean?

Line 221-222: “There were no differences between the time of in vitro maturation (24h vs 30h) in the percentage of isolated and matured oocytes” How did you reach this conclusion?

Line 223: Where this “(expended COC 2 vs 6.7)” come from? From the oocytes that were classified as compacted before IVM? Or from the expanded ones ?.

Author Response

Dear Reviewer 2

Thank you very much for your insightful review, which will improve the quality of our article.

I have attached a file below.

Anna M. Duszewska

Reviewer 3 Report

Introduction

The initial part of the introduction is very interesting, but really out of context. Hence, it should be reduced.

In contrast, the paragraph about cryoconservation of germplasm must be extended.

Materials and methods

2.1. Please described the procedures for selection of females. Also, please give a table with details of these animals (age, no. of gestations etc.)

2.3. The material taken from the epididymis of the male animal was not semen, please correct.

2.8. Please present evidence of normal distribution of data. Otherwise, please redo the analysis by using non-parametric tests.

Discussion

The discussion really needs extensive reshaping.

Whilst the findings are interesting, the discussion is not strong. Therefore, it needs rewriting.

I suggest dividing it in two or three subsections to make it easier for readers. Also, the strong points of the findings must be emphasised in the discussion.

Finally, there are some relevant references in the international literature, which have been omitted and should be cited.

General. The paper can become publishable after extensive revision, with special emphasis in rewriting the discussion.

Re-evaluation is necessary.

Author Response

Dear Reviewer

Thank you very much for your insightful review, which will improve the quality of our article.

Regarding your suggestions:

1. Introduction

a-The initial part of the introduction is very interesting, but really out of context. Hence, it should be reduced.

We did it

b - In contrast, the paragraph about cryopreservation of germplasm must be extended.

We did it

Materials and methods

2.1. Please described the procedures for selection of females. Also, please give a table with details of these animals (age, no. of gestations etc.)

In the case of wisent, it is not easy to talk about the production of embryos in vitro because each female is qualified for this procedure, as genetic material should be preserved from each (the same applies to males). Therefore, this procedure is adequate with assisted reproductive technology (ART) used in humans than in vitro embryo production (in this case, only the best females are qualified).

The females in this study were 3-11 years old and were not pregnant.

We added this information in the text (lines: 152-153)

2.3. The material taken from the epididymis of the male animal was not semen, please correct.

We did it

2.8. Please present evidence of normal distribution of data. Otherwise, please redo the analysis by using non-parametric tests.

In most of our statistical analysis, we used non-parametric tests. However, we verified other and changed statistic tests on non-parametric tests.

5.Discussion

5a.The discussion really needs extensive reshaping.

  1. Whilst the findings are interesting, the discussion is not strong. Therefore, it needs rewriting.
  2. I suggest dividing it in two or three subsections to make it easier for readers. Also, the strong points of the findings must be emphasised in the discussion.

We did it.

Also, we verified and added others' articles, but if you suggest the same others, we are open to putting them.

Thank you very much for your engagement to help improve our article.

Sincerely,                          

Anna M. Duszewska

Round 2

Reviewer 2 Report

This is a re-submission of a previous manuscript. Although the authors have done some modifications according to the reviewers' suggestions, this manuscript can not be accepted in the present version and some major points still should be addressed.
The authors concluded that 30h of IVM is better than 24 h because they did not obtain a single morula/early blastocyst stage embryo by fertilizing oocytes that matured in vitro for 24 hours. However, they obtained one morula or early blastocyst stage embryo from each donor's oocyte, which matured for 30 hours.  In my previous review, I mentioned that this conclusion couldn't be withdrawn because the initial conditions for both treatments (24 and 30h) were not the same. The answer of the authors was "This is evidence that this extension of oocyte maturation time was the limiting factor in this system as the other conditions were the same."  and " it is worth emphasizing that this did not influence the conclusions because the mean number of nacked and compact COC was the same in the case of oocytes matured for 24 and 30 h". I realize that all IVM, IVF, and IVC conditions were the same but not the quality of oocytes. As shown in Table 1, there are less compacted, more expanded, and less denuded COCs at 0 hours in the 24 h group than in the 30 h group, So, the quality of the total number of oocytes at 0 h ( germinal vesicle) in vitro matured for 24 and 30 h was not the same.  And to me, denuded and expanded oocytes shouldn't have been in vitro matured.
I do not also agree with the sentence  "There were no differences between the time of in vitro maturation (24h vs 30h) in the percentage of matured oocytes" (L236-217). Nor with the results for immature COCs and matured COCs in figure 1. To me, the percentage of in vitro matured oocytes can't be measured as the ratio of the number of useable COC (compact, expanded, and denuded COCs) to the number of recovered COC (compact, expanded, denuded, and degenerated COCs). After IVM some oocytes from each category may have not reached the maturation stage. So, nuclear maturation rate has to be assessed somehow (polar body extrusion, staining of the nucleus,… etc). I haven't worked in wisent species but it is not difficult to identify the first polar body in bovine oocytes under a stereomicroscope. It is placed between the oocyte and the zona pellucida and it has nothing to do with the high lipid content of the oocytes. 
The authors also mention that they had no donors and I understand the problem. But the lack of donors does not mean to reach incorrect conclusions.

Author Response

Dear Reviewer

We are writing to inform you that we have improved our article. However, we are conscious that you have other opinions about the qualification of oocytes for maturation and fertilisation. Therefore, please, consider the situation in which we would like to establish an immature and matured oocytes bank (this situation has been placed in human ART), and we must use only!!! necked oocytes. So this study gives us an opinion about using natural naked oocytes and their competencies.

Moreover,  in our opinion, it is impossible to see under a light microscope the first polar body in matured oocytes and pronuclei in zygotes. However, I know that is not a problem in some mammals, including mice, humans, and rabbits (I have worked with these species, so I have many experiences and……….knowledge!!!).

Thank you very much for your engagement.

Sincerely,                         

Anna M. Duszewska

Reviewer 3 Report

The manuscript has been improved.

Please undertake a thorough and extensive correction of linguistic matters to reshape the grammar and vocabulary of the text, as there are many lapses (probably inadverted, but still evident) scattered throughout the text.
Then it can be accepted.

Author Response

Dear Reviewer

Thank you very much for your engagement to help improve our article.

Regarding your suggestion, we corrected our article according to the MDPI's English editing service (attachment).

Sincerely,                         

Anna M. Duszewska
